# DEFENDING AGAINST RECONSTRUCTION ATTACKS WITH RÉNYI DIFFERENTIAL PRIVACY

## ABSTRACT

Reconstruction attacks allow an adversary to regenerate data samples of the training set using access to only a trained model. It has been recently shown that simple heuristics can reconstruct data samples from language models, making this threat scenario an important aspect of model release. Differential privacy is a known solution to such attacks, but it is often used with a large privacy budget ($\varepsilon \geq 8$) which does not translate to meaningful guarantees. Our main contribution is stronger privacy guarantees against *reconstruction attacks* improving on existing literature. In particular, we show that larger privacy budgets do not provably protect against membership inference, but can still protect against extraction of high-entropy secrets. We design a method to efficiently run reconstruction attacks with lazy sampling and empirically show that we can expose at-risk training samples from non-private language models. We show experimentally that our guarantees hold on language models of practical significance trained with differential privacy, including GPT-2 finetuned on Wikitext-103.

## 1 INTRODUCTION

Probabilistic generative models are trained to assign high likelihood to data from the training set. In the case of language models, the decomposition $\mathbb{P}(x_1, \ldots, x_T) = \Pi_{i=1}^{T} \mathbb{P}(x_i \mid x_{<i})$ also allows for efficient sampling of sequences. Given such models will *overfit* on the training set data, sampling from a trained model will sometimes yield verbatim sentences from the training set. Carlini et al. (2021b) leverage this effect along with clever filtering techniques to produce samples that are likely to be in the training set. Their work demonstrates that *reconstruction attacks* are not only possible on large-scale generative language models such as GPT-2 (Radford et al., 2019) but also successful: their best attack reaches 66% precision on the top-100 generated sentences. Another category of attacks is *membership inference*, where the adversary has access to both the trained model and a data sample, and has to predict whether this data sample comes from the training set. This attack is easier because the task of the adversary is simpler.

The standard defense against such privacy attacks is differential privacy (DP) (Dwork et al., 2006; Dwork & Roth, 2014). DP defines a privacy budget $\varepsilon$ that can be used to control the privacy/utility tradeoff of the trained model. However, there is no consensus over acceptable values of $\varepsilon$ and, in practice, $\varepsilon$ is often chosen to defeat practical membership inference attacks (Watson et al., 2021; Carlini et al., 2021a). In this paper, we show that Rényi differential privacy (RDP) (Mironov, 2017) can actually provide guarantees regarding the probability of reconstructing a sample; these guarantees are stronger than existing ones *for the same mechanism*. In particular, we show that there is an intermediate regime in which membership inference is not protected but full reconstruction of sufficiently high-entropy secrets remains difficult. This means that an adversary *who knows a secret $s$* can determine if it was in the training set, but extracting it from the model stays hard if they *do not know the secret $s$* in the first place.

We refer to samples that an adversary tries to reconstruct as *secrets*. Not all samples from the training set would be considered "secret" in the sense that their content is public knowledge, such as newspaper headlines. We circumvent this issue by considering all samples secrets, and quantifying their level of secrecy by the number of unknown bits of information. Specifically, given a probabilistic reconstruction model $\mathcal{A}$ that generates secrets from a trained model $\theta$, $s \sim \mathcal{A}(\theta)$, the secrecy of a sample $s$ is $b \triangleq \log_2 (1/\mathbb{P}(\mathcal{A}(\theta) = s))$. An adversary needs on average $2^b$ trials to chance upon the

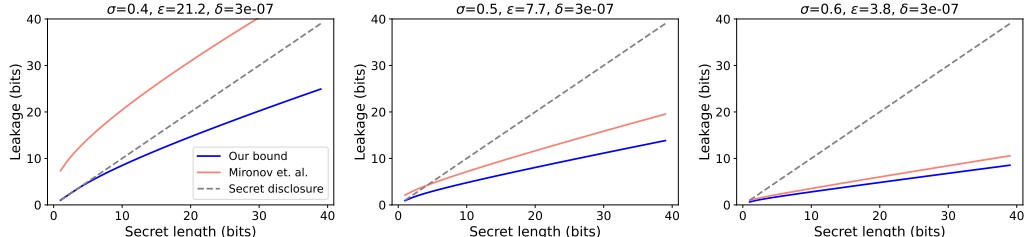

Figure 1: We train a neural network using DP-SGD and consider various levels of DP-noise $\sigma$ (increasing from left to right). We consider a secret of varying length (in bits $b$, $x$-axis) in the dataset and wish to estimate the number of bits from this secret that will leak after training ($y$-axis). For each level of DP-noise, we plot the upper bound $\min_\alpha h(\alpha, p_0)$ computed using Mironov et al. (2019) (see Equation equation 2) as well as our theoretical bound $L(p_0)$ against the secret length $b$ by setting $p_0 = 2^{-b}$. We additionally plot the line $y = x$: points below this line indicate that the secret will not entirely leak, points above means total leakage. To generate these plots, we use the settings of our real-life canary experiments in Section 4.3, with $186$k training steps and a sampling rate $q = 2.81 \times 10^{-4}$. We also report the privacy budget $\varepsilon$ at $\delta = 3 \times 10^{-7}$ in the plot titles computed using Balle et al. (2020), the standard practice with DP training. The plots demonstrate that our guarantee prevents from total secret disclosure for various levels of DP noise $\sigma$. Furthermore, we observe that our bound is comparatively better for lower levels of the DP noise $\sigma$.

secret $s$. This number of trials corresponds to the verification cost incurred by an adversary in many practical scenarios. For example if the adversary is guessing a phone number, they have to actually dial this number to "exploit" the secret.

A secret can have a non-zero probability even for a model that was not trained on this secret. As an extreme example, a language model generating uniformly random numbers will predict a 10-digit phone number with probability $10^{-10}$. The goal of a privacy-preserving training procedure is to ensure that the secret is not much more probable under a model that was trained on it. The length of the secret can vary depending on prior information: for phone numbers, knowing the area code reduces the secret from 10 to 7 unknown digits. Fortunately, RDP guarantees work against *any* prior knowledge, thanks to the post-processing property detailed in Section 2.2.

Our contributions are the following:

- We use the probability preservation guarantee from RDP to derive better secret protection. In particular, we show that the length of the secret itself provides more privacy.
- We empirically estimate the leakage for $n$-gram models and show that it matches our bound.
- We design a method to efficiently run the reconstruction attack of Carlini et al. (2021b) using lazy sampling and use it to surface at-risk samples on non-private language models.
- We fine-tune language models to competitive perplexity with differential privacy and show that the observed leakage under our attack model is smaller than the guarantee, even when we favor the adversary by increasing the secret's sampling rate.

## 2 BACKGROUND

Throughout the paper, we use $\log$ to denote the natural logarithm and $\log_2$ for the base 2 logarithm.

### 2.1 PRIVACY ATTACKS

**Membership Inference** attacks (Homer et al., 2008; Shokri et al., 2017) determine, given a trained model and a data sample, whether the sample was part of the model's training set. Given a sample $z$ and a model $\theta$ trained on a dataset $D$, the attacker's objective is to design a score function $\phi(\theta, z)$ such that $\phi$ is high when $z \in D$ and low otherwise. Various score functions have been proposed, such as the gradient norm (Nasr et al., 2019), the model confidence (Salem et al., 2018), or an output of a neural network (Shokri et al., 2017). Surprisingly, choosing the score function to be the loss $\mathcal{L}(\theta, x)$

is a simple and effective approach (Salem et al., 2018; Sablayrolles et al., 2019). Recent works argue that a practically relevant threat model of membership inference is to confidently predict training set membership of a few samples rather than guessing well on average (Watson et al., 2021; Carlini et al., 2021a; Ye et al., 2021). Such membership inference attacks are evaluated using true and false positive rates. In this context, calibrating the loss by centering (Watson et al., 2021) or by fitting a Gaussian likelihood (Carlini et al., 2021a) further improves the performance of the attack and yields state-of-the-art results. This kind of attacks essentially identifies uncommon samples on which a model is overconfident. These samples are also the focus of our study.

**Language Model Extraction.**    Carlini et al. (2019) propose a methodology to quantify the exposure of unique and secret training sequences to such extraction attacks. To this end, the authors insert a *canary* in the training set and measure its exposure as the excess belief that the model has in the canary over random chance. The authors then conclude that differential privacy is a suitable mitigation technique, although at the cost of some utility. Similarly, Carlini et al. (2021b) show that it is possible to extract hundreds of training sentences in a two-step procedure when attacking GPT-2, a language model trained on large scrapes of the public internet (Radford et al., 2019). First, the authors generate 200,000 samples for each of the following sampling strategies: sampling with a linearly decaying temperature, top-$k$ sampling and sampling conditionally on random Internet text. Then, they reduce the problem to Membership Inference among the generated samples and select the top-100 samples most likely to be members according to different filtering strategies. Depending on the metrics to rank the generated sentences (loss of the target model calibrated or not with a loss of a smaller model for instance), the authors identify a few dozens of training sentences from the top-100 sentences. Finally, a recent line of work focuses on providing differentially private predictions based on a pretrained, non-DP model (Ginart et al., 2022; Majmudar et al., 2022).

## 2.2 DIFFERENTIAL PRIVACY

Differential Privacy is a standard for privacy guarantees (Dwork et al., 2006; Dwork & Roth, 2014).

**Definition 1.** *A randomized mechanism $\mathcal{M}\colon \mathcal{D} \to \mathcal{R}$ satisfies $(\varepsilon, \delta)$-differential privacy (DP) if, for any adjacent inputs $D, D' \in \mathcal{D}$ and for any $S \subset \mathcal{R}$, we have $\mathbb{P}[\mathcal{M}(D) \in S] \leq e^{\varepsilon}\mathbb{P}[\mathcal{M}(D') \in S] + \delta$.*

Datasets $D$ and $D'$ are *adjacent* if they differ by at most one element. Rényi Differential Privacy (RDP) (Mironov, 2017) was introduced to obtain tighter composition properties. We recall here the properties of RDP that will be useful for the rest of the paper, while referring the reader to Mironov (2017) for a more comprehensive overview.

**Definition 2.** *For two probability distributions $P$ and $Q$ defined over $\mathcal{R}$, the Rényi divergence of order $\alpha > 1$ is*

$$D_\alpha(P \parallel Q) \triangleq \frac{1}{\alpha - 1} \log \mathbb{E}_{x \sim Q} \left( \frac{P(x)}{Q(x)} \right)^\alpha .$$

**Definition 3.** *A randomized mechanism $\mathcal{M}\colon \mathcal{D} \to \mathcal{R}$ satisfies $(\alpha, d_\alpha)$-Rényi differential privacy (RDP) if, for any adjacent inputs $D, D' \in \mathcal{D}$, we have*

$$D_\alpha(\mathcal{M}(D) \parallel \mathcal{M}(D')) \leq d_\alpha.$$

RDP guarantees can be converted into DP guarantees while the converse is not true, making RDP a strictly stronger property. If $\mathcal{M}$ is $(\alpha, d_\alpha)$-RDP, then it is also $\left( d_\alpha + \frac{\log 1/\delta}{\alpha - 1}, \delta \right)$-DP for any $0 < \delta < 1$. This conversion to $(\epsilon, \delta)$-DP is slightly improved by Balle et al. (2020).

**Properties of RDP.**    As with $(\varepsilon, \delta)$-DP, $(\alpha, d_\alpha)$-RDP guarantees are preserved by *post-processing*: if $\mathcal{M}$ is $(\alpha, d_\alpha)$-RDP and if $\mathcal{A}\colon \mathcal{R} \to \mathcal{R}'$ is a randomized mechanism, then $\mathcal{A} \circ \mathcal{M}$ is $(\alpha, d_\alpha)$-RDP.

**Probability Preservation.**    A direct consequence of $(\alpha, d_\alpha)$-RDP is quantified by the inequality (Mironov, 2017):

$$e^{-d_\alpha} p^{\alpha/(\alpha-1)} \leq p' \leq \left( e^{d_\alpha} p \right)^{(\alpha-1)/\alpha} \tag{1}$$

where $p \triangleq \mathbb{P}[\mathcal{M}(D) \in S]$ and $p' \triangleq \mathbb{P}[\mathcal{M}(D') \in S]$. Informally, since $\mathcal{M}(D)$ and $\mathcal{M}(D')$ are close, the probabilities of any event $S$ under both $\mathcal{M}(D)$ and $\mathcal{M}(D')$ are also close.

**DP-SGD.** Private training of neural networks is usually conducted with DP-SGD (Abadi et al., 2016; Song et al., 2013; Bassily et al., 2014) as follows. For each training step $t$, we gather a batch (of average length $L$) of training examples $z^{(i)}$ by sampling each element without replacement with probability (or sampling rate) $q$. We compute the per-sample gradients $g_t(z_i)$, clip them to a constant $C > 0$, average them and add Gaussian noise with variance $\sigma^2 C^2$:

$$\bar{g}_t(z_i) = g_t(z_i)/\max(1, \|g_t(z_i)\|_2/C), \qquad \tilde{g}_t = \frac{1}{L}\left(\sum_i \bar{g}_t(z_i) + \mathcal{N}\left(0, \sigma^2 C^2\right)\right),$$

and we use the noisy gradient $\tilde{g}_t$ in the optimization step. Finally, an *accountant* tracks the Rényi privacy $d_\alpha$ over all the training steps. This quantity only depends on the sampling rate $q$, the number of training steps and the noise multiplier $\sigma$.

**NLP with Differential Privacy.** Recent work has shown that fine-tuning language models to competitive accuracy with differential privacy is possible. For instance, Li et al. (2021) provide a recipe to directly fine-tune large transformer models with 50 to 300 million parameters directly with DP at privacy level $\varepsilon \in \{3, 8\}$ for various downstream tasks. Similarly, Yu et al. (2021) finetune privately competitive RoBERTa-Large models (Liu et al., 2019) with $\varepsilon = 6.7$ by training only a fraction of the network's parameters using low-rank weight matrices or skip-connections.

## 3 OUR METHOD

We first derive an upper bound on the information leakage, and show that it provides better privacy guarantees compared to traditional ones (Mironov, 2017). Then, we present a *lazy* method to efficiently identify samples that are likely to leak when performing a reconstruction attack.

### 3.1 RECONSTRUCTION AND CANARIES

The goal of reconstruction attacks is to surface training samples given access to the target network's weights. If $D$ is a dataset and $s$ a secret, we denote by $D' = D \cup \{s\}$. Recall that $\mathcal{M}(D)$ (resp., $\mathcal{M}(D')$) represents the distribution of the target network's weights after training on $D$ (resp., $D'$). We assume Finally, $\mathcal{A}$ denotes the attack mechanism that takes the target network's weights and outputs a probability distribution over secrets. For a given secret s, we note $p_0 \triangleq \mathbb{P}[\mathcal{A}(\mathcal{M}(D)) = s]$ and $p_1 \triangleq \mathbb{P}[\mathcal{A}(\mathcal{M}(D')) = s]$. We thus sometimes refer to $p_0$ as the prior (i.e. the secrecy of $s$ before $s$ is added to dataset $D$), and $p_1$ as the posterior.

**Theorem 1.** *If $\mathcal{M}$ is $(\alpha, d_\alpha)$-RDP, the leakage $\log(p_1/p_0)$ is bounded by*

$$\underbrace{-d_\alpha - \frac{\log(1/p_0)}{\alpha - 1}}_{\triangleq -h(\alpha, p_0)} \leq \log\left(\frac{p_1}{p_0}\right) \leq \underbrace{d_\alpha \frac{\alpha - 1}{\alpha} + \frac{\log(1/p_0)}{\alpha}}_{\triangleq l(\alpha, p_0)}. \tag{2}$$

*Finally, since $\alpha > \alpha - 1$, we have $l(\alpha, p_0) < h(\alpha, p_0)$.*

*Proof.* Thanks to post-processing, $\mathcal{A} \circ \mathcal{M}$ is $(\alpha, d_\alpha)$-RDP, and using the Probability Preservation inequality equation 1 gives the result. See Appendix A.1 for intermediate calculations. □

We emphasize that Theorem 1 applies to *any* attack mechanism $\mathcal{A}$, as $\mathcal{A} \circ \mathcal{M}$ is RDP as long as $\mathcal{M}$ is.

**Corollary 1.** *Since the leakage $\log(p_1/p_0)$ is independent of $\alpha$, we strengthen the bound of Theorem 1 by minimizing over orders:*

$$\log\left(\frac{p_1}{p_0}\right) \leq L(p_0) \triangleq \min_{\alpha > 1} l(\alpha, p_0). \tag{3}$$

**Comparison with traditional DP guarantees.** Theorem equation 1 implies a bound on the absolute leakage as follows:

$$\left|\log\left(\frac{p_1}{p_0}\right)\right| \leq \max(l(\alpha, p_0), h(\alpha, p_0)) = h(\alpha, p_0).$$

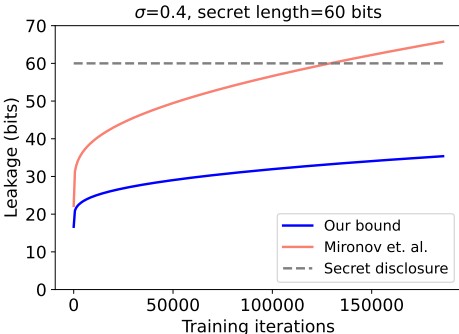

Figure 2: We consider training a neural network with DP-SGD with the setup described in Figure 1. Here, we fix the secret length to $b = 60$ bits and plot the dependency of the leakage with the number of training iterations as long as our empirical measurement (see Figure 3). Interpretation: even when repeating the canary multiple times, the empirical observed leakage in a real-life scenario is still lower than our predicted bound $L(p_0)$.

If we take $\delta = p_0$, this corresponds to the traditional $\varepsilon$ given by $\min_\alpha h(\alpha, \delta)$. Given that $l(\alpha, p_0) < h(\alpha, p_0)$, our bound on leakage is *better* than traditional guarantees from $(\varepsilon, \delta)$-DP. These values are shown in Figures 1 and 2: for lower values of the noise multiplier $\sigma$, the bound provided by $l$ is much lower than $h$, while the gap between the two decreases as the noise multiplier $\sigma$ gets bigger. We can extend that analogy and compare our leakage guarantee to the privacy budget $\varepsilon$ that would be given for a probability of failure $\delta = p_0$. Numerically, our bounds are also better than the slightly tighter bounds by Balle et al. (2020) as depicted in Appendix A.3, although not directly comparable.

**Absolute Leakage.** Nominally, a bound on the absolute leakage is a stronger guarantee, because it also prevents the posterior $p_1$ from becoming *smaller* than the prior $p_0$. However, we argue that this case is much less relevant from the risk perspective: even if the probability of other outcomes $s'$ becomes less likely, "mechanically" increasing the probability of the secret $s$, the probability of the secret is still bounded by the right-hand side of Equation equation 2. Furthermore, we argue that negative membership inference, *i.e.*, predicting that a sample was *not* part of a training set, is not as concerning as positive membership inference. Indeed, an individual can be present in a data collection pool, but not included in a particular training set for a variety of reasons, hence absence from the training set does not imply absence from the data collection stage.

**Length of the Secret.** With a slight abuse of notation, let us consider the leakage (in bits) as a function of the number of secret bits $L_2(b) \triangleq \min_\alpha d_\alpha \frac{\alpha - 1}{\alpha \log(2)} + \frac{b}{\alpha}$. Our leakage function $b \mapsto L_2(b)$ satisfies two properties: it is non-decreasing and concave (see Appendix A.2 for proof). Thus, longer secrets will leak more in absolute terms, but less in relative terms. Let us assume that we are looking at a particular secret of binary nature (such as whether some individual owns a particular object for example), with prior $p_0 = 2^{-b}$. Even though there are two possible outcomes, the prior $p_0$ is not necessarily equal to $1/2$: if the item is rare, the prior is more likely to be smaller. The log-posterior $\log_2(p_1) < -b + L_2(b)$, a non-increasing function (see Appendix A.2 for proof). This upper-bound is maximized when $b = 0$: longer secrets lead to smaller values of $p_1$. The most sensitive secrets are the ones that are the most likely in the first place, and the length of the secret itself acts as a protection against reconstruction attacks.

**Comparison to membership inference.** In particular, membership inference attacks correspond to attacks with a low number of secret bits. These attacks are usually conducted with a prior probability of $1/2$ (Yeom et al., 2018; Sablayrolles et al., 2019), and even though some works consider different ratios of positive to negative members (Watson et al., 2021; Rezaei & Liu, 2021), they stay within a factor of 1 to 10. Some privacy settings (noise level $\sigma$, sampling rate $q$ and number of steps) will thus offer no guarantee against membership inference (because of the low number of bits to guess) but still will not allow for full reconstruction of the rarest samples (high number of bits).

## 3.2 Lazy Sampling: Identifying Samples Likely to Leak

Let us assume that we want to generate a secret $s$ from a model $\theta$, using the probabilistic attack model $\mathcal{A}(\theta)$. The method of Carlini et al. (2021b) requires sampling hundreds of thousands of times from $\mathcal{A}(\theta)$ in the hope that we find the secret $s$ (and then filter out the generated samples using calibrated membership inference scores). Fortunately, most attacks of Carlini et al. (2021b) are amenable to *lazy sampling*: we can take a secret $s$, and directly compute its probability $p = \mathbb{P}(\mathcal{A}(\theta) = s)$.

Not all probabilistic processes $\mathcal{A}$ have a tractable density $\mathbb{P}(\mathcal{A}(\theta) = s)$. For instance, generating a sequence $x_1, \ldots, x_T$ and only keeping the last samples $x_i, \ldots, x_T$ does not have a tractable density. On the other hand, vanilla sampling from a language model $\theta$ can be done lazily because the probability of a sequence $x_1, \ldots, x_T$ can be expressed as $\Pi_{i=1}^{T} f_{\theta,i}(x_i \mid x_{1:i})$ where $x_{1:i} = (x_1, \ldots, x_{i-1})$ and $f_{\theta,i}(x_i \mid x_{1:i})$ is the conditional probability of $x_i$ given the past. While this is straightforward for regular sampling, it is also true for temperature and top-$k$ sampling, with the caveat that these probabilities can sometimes be 0, as illustrated in Appendix A.5.

We define $T_k(\theta, x_{1:i})$ the set of top-$k$ predictions and $\beta_1, \ldots, \beta_T$ a set of temperatures. From there, we can define a top-$k$ and/or temperature language model as

$$\lambda(x_i \mid x_{1:i}) \triangleq \begin{cases} \dfrac{f_\theta(x_i | x_{1:i})^{\frac{1}{\beta_i}}}{\sum_{y \in T_k(\theta | x_{1:i})} f_\theta(y | x_{1:i})^{\frac{1}{\beta_i}}} & \text{if } x_i \in T_k(x_{1:i}) \\ 0 & \text{otherwise.} \end{cases} \qquad \lambda(x_1, \ldots, x_T) = \Pi_{i=1}^{T} \lambda(x_i \mid x_{1:i}).$$

Lazy sampling allows us to analyze two of the main strategies of Carlini et al. (2021b), but does not apply to the Internet sampling one. Indeed, Internet sampling first chooses a text $c$ crawled from the public web, and generates iteratively from it $f_\theta(\cdot, c)$, effectively using $c$ as a prompt.

**Leakage approximation.** In the remainder of this paper (except for Section 4.1), we make the following assumption:

**Assumption 1.** *For any $\theta \sim \mathcal{M}(D)$ and $\theta' \sim \mathcal{M}(D)$,*

$$\log \mathbb{P}(\mathcal{A}(\theta) = s) \approx \log \mathbb{P}(\mathcal{A}(\theta') = s).$$

To support Assumption 1, we measured the mean and standard deviation of the log-probability of the secret $\log(p_1)$ for our $n$-gram model detailed in Section 4.1. For each level of DP noise $\sigma$, we consider 10,000 models and observe the probability of leakage is tightly concentrated around the mean. For instance, with $\sigma = 2.875$, $\log p_1$ has mean $\simeq -22.5$ and standard deviation $\simeq 1.5$, from which we conclude that the log-probability of a secret does not vary much when re-training a model on a fixed dataset $D$.

Using Assumption 1 and lazy sampling, we can take all sentences $s = x_1, \ldots, x_T$ from the training set, and compute their probability of being generated by the attack $\mathcal{A}$, and compare it to their "score" that is used at the filtering stage. The above strategy allows us to identify samples that would be reconstructed using this particular attack. Of course, if this particular attack fails to reconstruct some sample $s$, it does not mean that all attacks would fail on $s$.

## 4 Experiments

We first experiment with a simple private $n$-gram language model to empirically compare the secret leakage with our bounds. We then consider a non-private language model that exhibits samples that are the most at risk for a reconstruction attack using the lazy strategy for the Carlini et al. (2021b) attack. Finally, we experiment on private fine-tuning of GPT-2, with parameters yielding low perplexity (Li et al., 2021), and show that the empirical leakage is even smaller than predicted by our bound. Overall our experiments required in the order of 200 GPU-days on V100 GPUs.

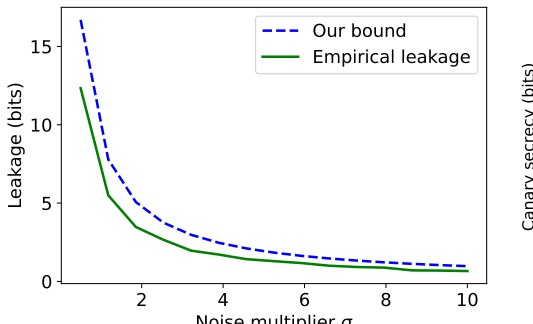 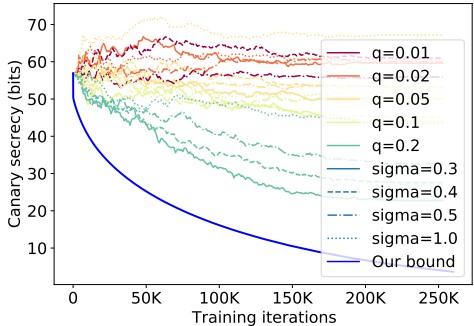

Figure 3: **Left**: Empirical leakage $\log(p_1/p_0)$ computed using Equation 4 on $n$-gram language models, compared to our theoretical bound $L(p_0)$ from Corollary 1. We can see that our bound is tight. **Right**: We empirically measure the leakage of our canary (a fixed 16-digits random number) when training our GPT-2 model on Wikitext with DP. If the model predicts the canary with a given probability $p_1$, we compute the secrecy $b_1 = \log_2(1/p_1)$ ($y$-axis, in bits). The blue curve corresponds to our bound for $\sigma = 1$ and a canary sampling rate $q = 1\%$. Empirically, the observed privacy is more than predicted, even for lower levels of DP noise $\sigma$. To make our case stronger, we measure canary secrecy with increasing canary sampling rates $q$ and show that our bound is verified.

## 4.1 $n$-GRAM LANGUAGE MODEL

We first experiment on a $n$-gram language model to compare our guarantees against empirical secret leakage. We consider $T > 0$ time steps and generate random digits $c \in \{0, 1, \ldots, 9\}^T$ (the *canary*). Our objective is to train the $n$-gram language model on the (fixed) canary $c$ only.

A full $n$-gram language model $f_\theta$ has $\sum_{i=1}^T 10^i \approx 10^T$ parameters. However, the only parameters of interest in our case are the ones corresponding to $f_\theta(c_i|c_{1:i})$. These parameters suffice to compute the probability of the sequence $c$ according to the model $f_\theta$, and the rest of the parameters correspond to other "branches" of the language model. Hence, we can train $f_\theta$ lazily by only modifying these parameters, which brings the number of parameters down to $10T$. We train our model with the softmax loss. More specifically, our loss $\mathcal{L}$ writes

$$\mathcal{L}(\theta) = -\sum_{t=0}^{T-1} \log(u_{t,c_t}), \qquad u_{t,i} = \frac{e^{\theta_{t,i}}}{\sum_{d=0}^{D-1} e^{\theta_{t,d}}},$$

where $c_t$ is the digit at index $t$ in the canary and $u_{t,i}$ is the probability that digit $i$ appears at position $t$.

We experiment with Opacus (Yousefpour et al., 2021), using a randomly generated and fixed canary of length $T = 10$. We set the sampling rate to $q = 1$, the clipping factor $C = 1$, the DP noise level $\sigma \in [0.5, 10]$ and learning rate $\eta = 0.5/\sigma$. We now wish to measure empirically the leakage $\log(p_1/p_0)$. We have $p_1 = \mathbb{P}(\mathcal{A}(\mathcal{M}(D')) = s) = \mathbb{E}_{\theta \sim \mathcal{M}(D')}(f_\theta(c))$. We approximate $p_1$ using Monte-Carlo sampling, by training $N = 10{,}000$ models and computing

$$\log(p_1) \approx \log\left(\frac{1}{N}\sum_{k=1}^N f_{\theta^{(k)}}(c)\right). \tag{4}$$

Given that the probabilities $f_{\theta^{(k)}}(c)$ can be quite small, we perform computations in the log-space for numerical stability. We have $p_0 = 10^{-T}$ since the problem is invariant under permutation of digits.

Finally, we use the `RDPAccountant` from Opacus to compute $d_\alpha$ for a range of orders $\alpha \in [1.01, 63]$ and compute $L(p_0)$ as in Equation equation 3. The results are shown in Figure 3, where the empirical leakage refers to $\log(p_1/p_0)$ and where our theoretical bound refers to $L(p_0)$. We observe that the bound is relatively tight for this simple $n$-gram language model.

## 4.2 LAZY RECONSTRUCTION WITHOUT DEFENSES

We now consider a vanilla target model trained without DP on the OpenWebText dataset (Gokaslan & Cohen, 2019) and want to identify samples from the training set that are most at risk in the event

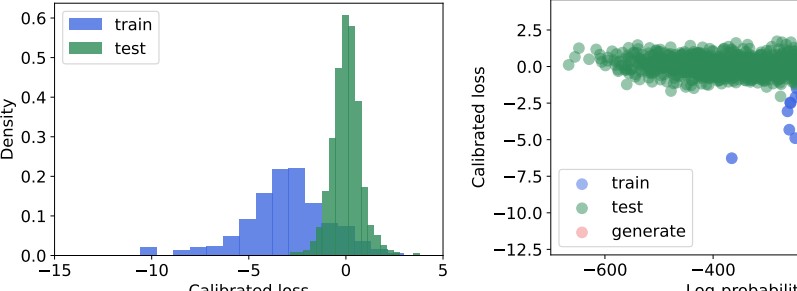

Figure 4: We train a vanilla language model without DP and compute, for a given sentence, its calibrated score using a model that is not trained on the same data, as well as its log-probability predicted by the target model. **Left:** We plot the histogram of a random subset of the train and test sets. **Right:** For a given sentence, we plot its calibrated loss ($y$-axis) against its log-probability of being generated by the target model ($x$-axis). In addition to sentences from the train and test set, we generate sentences from the target model and note that their average calibrated loss is lower than for sentences of the training set. The samples from the train set that are the most at-risk for a reconstruction attack are these with high log-probability (high probability of being generated by the attack model) and low calibrated loss (high probability of having been memorized by the target model). See Table 1 for selected sentences from the training set.

of a reconstruction attack. We compute, for a given sentence $z$, its calibrated score (or calibrated loss) defined as $\phi(\theta, z) = \mathcal{L}(\theta, x) - \mathcal{L}(\theta_0, x)$, where $\theta$ represents the target model's weights and $\theta_0$ represents a calibration model that is trained on held-out calibration data. We partition OpenWebText into training, calibration and test sets of 1 million samples each. The models are trained using the setup of Radford et al. (2019). As shown in Figure 4 on the left plot, we observe that samples from the training set have a lower calibrated loss, indicating that they have been memorized by the model. As expected, test samples have an average calibrated loss around 0 with a small standard deviation. On the right plot, we observe that the average calibrated loss for sentences generated by the target model is lower than for sentences from the training set. Given these measures, it is possible to identify samples that are most at risk due to a reconstruction attack: those are with high log-probability (higher odds of being generated by the target model) and low calibrated loss (high probability of being memorized by the target model).

Table 1 shows typical examples of elements from the training set with various levels of risk. Some of the sentences likely to be generated are indeed common (such as the two first rows). Other sentences however are very likely to be sampled (with a probability close to $10^{-3}$) and have a low calibration score, indicating they were memorized by the model. Finally, some sentences have a very low probability of being generated in the first place (less than 1 in 10 billions), so even though they are memorized by the model (low calibration score), they are unlikely to be surfaced. Carlini et al. (2021b) showed a somewhat analogous plot by displaying perplexity of the trained model against `zlib` entropy. There are two major differences: our $x$-axis shows the probability of being generated by the attack model (which is different from perplexity due to temperature and/or top-$k$ sampling), and our analysis is conducted on the *training* set, whereas their analysis is done on generated samples. In particular, we can have confidence that certain sentences with a very low probability of being generated will not be recovered *by this particular strategy*.

### 4.3 REAL-LIFE CANARY EXPERIMENTS

We also experiment with a realistic scenario of privately fine-tuning a large language model. We take a pre-trained GPT-2 model (Wolf et al., 2020) and fine-tune it on Wikitext-103 (Merity et al., 2017), available under the Creative Commons Attribution-ShareAlike License. We add a canary sentence to the training dataset and sample it during training with a sampling rate $q$. The canary consists of a prompt ("John Doe's credit card number is") and a secret, which is a string of 16 random digits. We are able to privately fine-tune GPT-2 with DP-SGD on Wikitext-103 and reach a perplexity of 45, which is very close to the non-private fine-tuning perplexity of 38 (refer to Appendix A.6 for details).

Table 1: We train a vanilla target model without DP on Wikitext-103 and display selected training sentences. A low calibrated loss denotes a sample that is likely to be memorized and a high probability denotes a sample that is more likely to be generated by the model. Hence, samples associated with a low calibrated loss and with a high generation probability are considered at risk. Note that $-11.3$ is quite low, as most generated samples will have a calibrated loss higher than $-7.5$ as in Figure 4.

| Sample | Probability | Calibration |
|---|---|---|
| *Advertisement* | $10^{-2.5}$ | $+0.10$ |
| *Content created by The Daily Caller News Foundation is available [...]* | $10^{-3.5}$ | $-0.18$ |
| *Subject: Games Day News and Rumours: Non Dark Eldar* | $10^{-3.6}$ | $-9.81$ |
| *Antarctic expeditioners honour traditional solstice swim* | $10^{-10}$ | $-11.3$ |
| *Homepage image courtesy of the Canadian Space Agency* | $10^{-21}$ | $+1.29$ |

Given the trained model, we approximate $p_1$ by computing the probability of the secret given the prompt $p_1 \approx f_\theta(\text{number} \mid \text{"John Doe's credit card number is"})$. With the chosen privacy parameters ($\sigma \in [0.3, 0.5]$, $q = 2.81 \times 10^{-4}$ and 186k steps), the empirical leakage is negligible. To make our case stronger, we increase the sampling rate of the canary in order to increase the empirical leakage, and observe that it is still lower than the proposed bound of Corollary 1.

**Results.** Figure 3 shows the empirical log probability of the secret as a function of the number of steps against the leakage guarantee computed according to Equation equation 3. With DP, decreasing levels of the noise $\sigma$ lead to increasing levels of leakage (or equivalently, decreasing levels of privacy). Similarly, increasing canary sampling rates $q$ result in more observed empirical leakage. We observe that even for the most favorable setup to canary memorization (low sigma, high sampling rate), the provided bound (computed for high sigma and low sampling rate) is verified. Finally, even with our least private version ($\sigma = 0.3$, $q = 0.2$), our strategy is not able to fully reconstruct the secret.

**Exposure.** Carlini et al. (2019) estimate the performance of their attack using the exposure metric, which is an approximation of the optimal number of guesses an adversary would need to correctly estimate a secret. In practice, we can only upper-bound the exposure because we have no guarantee that any attack is optimal. The canary secrecy showed in Figure 3 corresponds to exposure, but for the more recent reconstruction attacks of Carlini et al. (2021b).

## 4.4 LIMITS OF RECONSTRUCTION ATTACKS

Canary attacks (Carlini et al., 2019) are useful because the secret is chosen randomly and is thus independent of the rest of the dataset. In contrast, with practical reconstruction attacks it is difficult to estimate the randomness of the secret. Indeed, multiple identical sequences can be present in the dataset. Carlini et al. (2021b) correct for this by measuring $k$-eidetic memorization, *i.e.*, looking at sentences that appear $k$ times or less in the dataset. However, this does not account for knowledge shared across secrets. For example, if there is a secret in the form "[animal] drinks [beverage]", animal and beverage can vary in the dataset, so the language model will learn about this general structure during training: seeing "dog drinks water" can thus make "cat drinks milk" more likely, even if the latter does not appear in the dataset.

## 5 CONCLUSION

This work shows that Rényi Differential Privacy (DP-SGD) provides meaningful guarantees against reconstruction attacks. We also provide an efficient way to analyze the vulnerability of training samples to the reconstruction attack of Carlini et al. (2021b). Overall, the combination of our improved guarantees with the private fine-tuning of language models showcased by Li et al. (2021) allow us to train language models with a perplexity competitive with the state of the art and meaningful privacy guarantees against training data extraction. Finally, our work sheds light on the "higher information" end of the spectrum. First, reconstruction attacks are more credible because they only require access to a trained model and not to candidate samples. Second, as shown, reconstruction attacks can be defeated with levels of noise that would fail to defend against membership inference attacks. We hope that increased consideration for this threat model will drive adoption of DP-SGD as a standard in machine learning.

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

## A APPENDIX

### A.1 PROBABILITY PRESERVATION

Here we show the steps to get from Eq. 1 to Eq. 2:

$$\exp(-d_\alpha)p_0^{\alpha/(\alpha-1)} \le p_1 \le (\exp(d_\alpha)p_0)^{(\alpha-1)/\alpha}$$
$$-d_\alpha + \alpha/(\alpha-1)\log(p_0) \le \log(p_1) \le ((\alpha-1)/\alpha)(d_\alpha + \log p_0)$$
$$-d_\alpha + 1/(\alpha-1)\log(p_0) \le \log(p_1/p_0) \le (\alpha-1)d_\alpha/\alpha - 1/\alpha \log p_0.$$

### A.2 LEAKAGE FUNCTION

In this part, we prove that the leakage function $L_2$ is non-decreasing and concave. We also show that $\xi : b \mapsto L_2(b) - b$ is non-increasing. We start with two technical lemmas.

**Lemma 1.** *Given a family of non-decreasing functions $f_t, t \in \mathcal{T}$, the function $f(x) \triangleq \inf_t f_t(x)$ is non-decreasing.*

*Proof.* For $x < y$ and some $t \in \mathcal{T}$, we have

$$f(x) = \inf_{t'} f_{t'}(x) \le f_t(x)$$
$$\le f_t(y)$$

Since $f(x) \le f_t(y)$ for all $t$, we have $f(x) \le \inf_t f_t(y) = f(y)$. □

**Lemma 2.** *Given a family of concave functions $f_t, t \in \mathcal{T}$, the function $f(x) \triangleq \inf_t f_t(x)$ is concave.*

*Proof.* For a function $g$, we define its hypograph $\mathcal{H}(g) \triangleq \{(x, y) \mid y \le g(x)\}$. A function $g$ is concave iff its hypograph $\mathcal{H}(g)$ is a convex set. Each function $f_t$ being concave, its hypograph $\mathcal{H}(f_t)$ is a convex set. The hypograph $\mathcal{H}(f) = \cap_{t \in \mathcal{T}} \mathcal{H}(f_t)$ is an intersection of convex sets and is thus itself a convex set. Thus $f$ is concave. □

Now let us apply these lemmas to the leakage function. We have $L_2(b) = \min_\alpha f_\alpha(b)$ with each $f_\alpha(b) = d_\alpha \frac{\alpha-1}{\log(2)\alpha} + \frac{b}{\alpha}$. Each function $f_\alpha$ is linear, and hence concave, and non-decreasing because $1/\alpha > 0$. We can thus apply Lemma 2 and Lemma 1 to conclude that the leakage function $L_2$ is concave and non-decreasing.

The function $\xi : b \mapsto L_2(b) - b$ is a sum of a concave and a linear function and is thus concave. Thus, its derivative $\frac{\partial \xi}{\partial b}$ is non-increasing. Given that $\frac{\partial \xi}{\partial b}(0) = \frac{\partial L_2}{\partial b}(0) - 1 = 0$, $\frac{\partial \xi}{\partial b} \le 0$ and thus $\xi$ is non-increasing.

### A.3 COMPARISON TO BALLE ET AL. (2020)

In Figure 5, we display the numerical bounds of Balle et al. (2020).

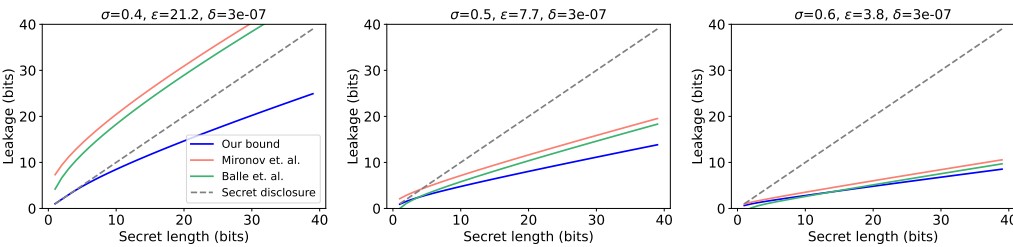

Figure 5: Comparison with Balle et al. (2020).

Figure 6 furthers the comparison to Balle with a different value of the sampling rate $q$.

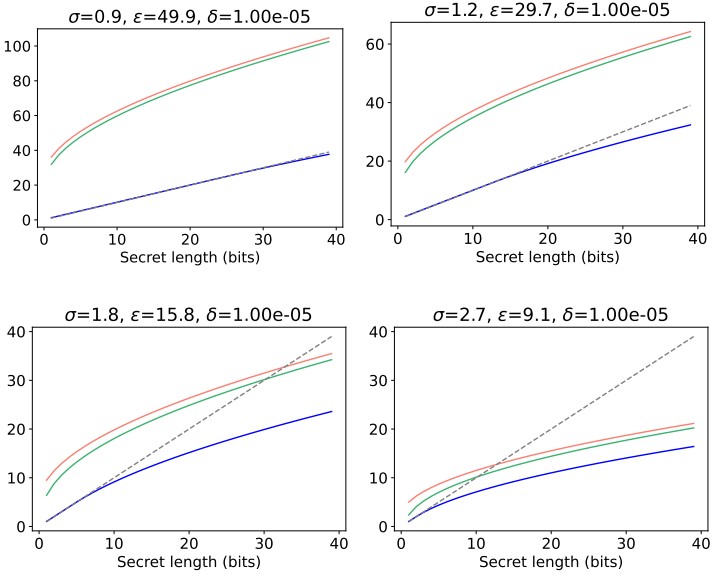

Figure 6: Comparison with Balle et al. (2020), on a DP-SGD run with $q = 20\%$ and 100 epochs, and various values of $\sigma$.

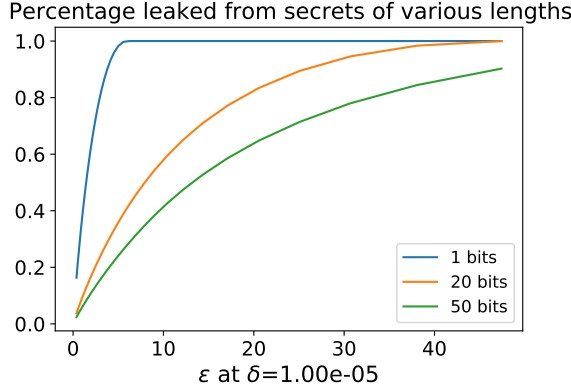

Figure 7: Percentage of secrets leaked as a function of privacy budget.

### A.4 COMPARISON OF MEMBERSHIP INFERENCE AND RECONSTRUCTION PRIVACY

Figure 7 shows the percentage of the secret leaked as a function of the privacy budget. This emphasizes that small, binary secrets leak completely for budgets $\epsilon < 10$, but longer secrets do not completely leak until much higher budgets (typically $\epsilon$ around 50 for 20-bit secrets).

### A.5 TOP-K SAMPLING

In Figure 8, we show that Top-$k$ sampling from a language model has a tractable density. The probability of a sequence is the product of the conditional probabilities along the path. If one word does not belong to the top-$k$ predictions, its probability will be 0, thus making the probability of the entire sentence 0.

### A.6 EXPERIMENTAL DETAILS

For WikiText-103, we follow the findings of Li et al. (2021) and use a large batch size ($B = 1024$), a small clipping threshold ($C = 1$), use AdamW (Loshchilov & Hutter, 2018) and freeze the embedding

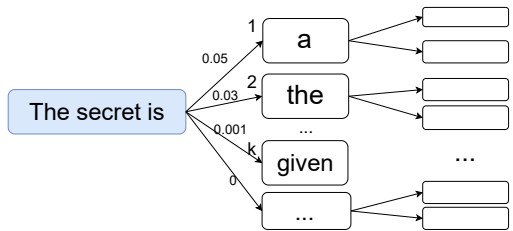

Figure 8: Top-$k$ sampling from a language model has a tractable density.

layers. We also find that using a low learning rate ($lr = 0.0001$) is crucial to avoid divergence. With these hyperparameters, we are able to fine-tune GPT-2 on Wikitext-103 and reach a perplexity of $45$, which is very close to the non-private fine-tuning perplexity of $38$ that we obtain. These results echo the findings of Li et al. (2021).

