# OpenReview forum: "Defending against Reconstruction attacks using Rényi Differential Privacy"
_ICLR.cc/2023/Conference — Submitted to ICLR 2023_

### Official Review · Reviewer_UcP3 · 2022-10-24

**Confidence:** 4
**Correctness:** 3
**Technical Novelty And Significance:** 3
**Empirical Novelty And Significance:** 2
**Recommendation:** 5

**Clarity, Quality, Novelty And Reproducibility:**

This framework is of average quality and kind of lacking innovation. The description of this framework is clear and the theoretical analysis is sufficient. In the experimental part, the validity analysis of the results is relatively redundant. The originality of the work is acceptable.

**Strength And Weaknesses:**

Strength
1. This paper considers the sample length and derives the privacy leakage boundary of training samples based on the Rényi differential privacy method, which seems to be reasonable and effective.
2. The authors propose a lazy-sampling-based method to improve the existing method to perform training data extraction attacks, which improves the performing efficiency and effectively identifies potentially private samples.
3. The organization and the results of the experiments in this paper appear to be efficient and reasonable.

Weaknesses
1. First, this article is difficult to read for me. I suggest that authors should be more careful with domain words and notations, such as “reconstruction attack”. In my background knowledge, a "reconstruction attack" in federated learning or graph learning for privacy-preserving is to reconstruct the training data using information (gradient or aggregated information) from the model training in the training process. However, I think that the "reconstruction attack" in this paper is an attack method in the inference process just like the member inference attack. I am confused if the “reconstruction attack” in this paper is the “training data extraction attack”. If it is a new attack, I would expect the author to redefine it.
2. In my opinion, the core idea of this paper is to use differential privacy to achieve that the generated samples using the random attack model are satisfying the privacy guarantee. Although the proposed method is effective and reasonable, it is incremental work and lacks innovation.
3. I think the organization of this paper needs improving. The framework of this paper looks fine, important sections are included and organized in a reasonable way. However, the content in each section is not easy to follow, and the organization of each section can be improved.

**Summary Of The Paper:**

This paper proposes a privacy protection method against training data extraction attacks from language model based on Rényi differential privacy. The authors point out the effect of sample length on privacy leakage and derive privacy bound based on Rényi differential privacy, and then propose an improved method based on lazy sampling to identify training samples that may leak privacy. The authors construct corresponding experiments for each part of the proposed method by fine-tuning the language model and show that the privacy leakage under the proposed attack model is less than the privacy guarantee.

**Summary Of The Review:**

This paper proposes a privacy protection method against training data extraction attacks from language model based on Rényi differential privacy. However, this work lacks some innovation for ICLR and the content in each section is not easy to follow and the organization can be improved.

---

> ### Author Response · Authors · 2022-11-18
> **Response to the reviewer**
>
> We thank the reviewer for their review and answer the questions below.
>
> > First, this article is difficult to read for me. I suggest that authors should be more careful with domain words and notations, such as “reconstruction attack”. In my background knowledge, a "reconstruction attack" in federated learning or graph learning for privacy-preserving is to reconstruct the training data using information (gradient or aggregated information) from the model training in the training process. However, I think that the "reconstruction attack" in this paper is an attack method in the inference process just like the member inference attack. I am confused if the “reconstruction attack” in this paper is the “training data extraction attack”. If it is a new attack, I would expect the author to redefine it.
>
> We thank the reviewer for pointing out the potential name conflict with other types of attacks. We are indeed concerned with training data extraction attacks, that we call "reconstruction attacks". We will clarify that in the paper to avoid confusion with other attacks that are also named reconstruction attacks.
>
> > In my opinion, the core idea of this paper is to use differential privacy to achieve that the generated samples using the random attack model are satisfying the privacy guarantee.
>
> Our paper shows indeed that using DP-SGD provides guarantees against reconstruction (training data extraction) attacks. We emphasize that our guarantees work against *any* attack model.
>
> > Although the proposed method is effective and reasonable, it is incremental work and lacks innovation.
>
> To the best of our knowledge, our paper is the first to show that using vanilla DP-SGD converts into protection against training data extraction. While DP is known to provide guarantees against membership-inference style attacks, there is no work that addresses reconstruction of training samples to the best of our knowledge. In particular, these membership inference style guarantees are independent of the secret lengths. In other words, the bounds for MI apply similarly to protect low-entropy information such as gender or high-entropy information such as credit card numbers, full name and address, etc. Our paper is the first to show, using guarantees from RDP, that longer secrets indeed enjoy better protection.
>
> > I think the organization of this paper needs improving. The framework of this paper looks fine, important sections are included and organized in a reasonable way. However, the content in each section is not easy to follow, and the organization of each section can be improved.
>
> We thank the reviewer for their feedback, we will try to improve the flow inside the sections.

---

### Official Review · Reviewer_ruMF · 2022-10-28

**Confidence:** 2
**Correctness:** 3
**Technical Novelty And Significance:** 2
**Empirical Novelty And Significance:** 2
**Recommendation:** 6

**Clarity, Quality, Novelty And Reproducibility:**

To me, the definition looks vague. Is secret defined as a full sample or a part of a sample?
The example discussed in the introduction looks like a  part of a sample, but from the definition in Section 3, it looks like a full sample. In experiments, the bit size of the secret is controlled. If the secret is a full sample, this cannot happen.

I could not well understand the relationship between Section 3.1 and Section 3.2.

Is it possible to characterize the absolute leakage concerning sample size or some other factors that we can control?


**Strength And Weaknesses:**

Strength
- Considered an interesting and novel problem
- The considered problem well fits privacy that should be considered in the real world

Weakness
- Characterization of the obtained result seems to be a bit weak


**Summary Of The Paper:**

This work examines whether the secret information that may be contained in the training data set can be extracted from a model which satisfies RDP. This problem is positioned as a problem of intermediate difficulty between the reconstruction and membership attacks. The authors define leakage as the probability that an attacker identifies a secret in a model learned from data that contains the secret and in a model learned from data that does not. Through analysis, the authors obtain upper bounds on the log ratio of the two and compare them to empirical evaluations.

**Summary Of The Review:**

Overall, the manuscript considers an interesting problem.
Experimental results are interesting, while the characterization of theoretical results seems to e a bit weak.

---

> ### Author Response · Authors · 2022-11-18
> **Response to the review**
>
> We thank the reviewer for their review and answer the questions below.
>
> > Is secret defined as a full sample or a part of a sample?
>
> Our definition of secret encompasses both full samples, part of samples, and side information. To state it more clearly, we consider an individual, *full* sample to be a secret but we assume an informed adversary (this adversary can already know some parts of the sample or make educated guesses through malicious prompts for instance). As an example, the secret can be the sentence "My credit card number is 765 789". If the adversary knows the prompt ("My credit card number is"), the secret has 6 digits of uncertainty (\~20 bits). If the adversary already knows (from side information), say, the first three digits then the secret only has 3 digits of uncertainty (\~10 bits).
>
> > In experiments, the bit size of the secret is controlled. If the secret is a full sample, this cannot happen.
>
> We do not control the bit size of the secret (because we allow the adversary side information), but for each bit size there is a corresponding guarantee.
>
> > Is it possible to characterize the absolute leakage concerning sample size or some other factors that we can control?
>
> To the best of our knowledge, DP protects individual samples so the sample size does not factor in the leakage.

---

> > ### Comment · Reviewer_ruMF · 2022-12-14
> > **Response to rebuttal**
> >
> > Thank you for your answers to my questions. I would like to keep my score the same, taking your answers into consideration.

---

### Official Review · Reviewer_2RkH · 2022-10-30

**Confidence:** 3
**Correctness:** 3
**Technical Novelty And Significance:** 2
**Empirical Novelty And Significance:** 3
**Recommendation:** 5

**Clarity, Quality, Novelty And Reproducibility:**

Clarity: Okay, can be understood, but require some effort.

Quality: Okay, there might be some technical issues as mentioned in the weakness.

Novelty: Okay, but not extremely novel

Reproducibility: N.A., I am not an expert on NLP. Thus, I have no idea whether the experiments can be reproduced or not.

**Strength And Weaknesses:**

Strength: The theoretical bound in this paper is much better than the state of the art when \sigma is small (e.g., 0.4). This makes the bound useful for application scenarios with moderate privacy requirements (but may with strong accuracy requirements)

Weakness:
1. The theoretical bound in this paper is just slightly better than the state of the art when \sigma becomes larger (e.g., 0.5), which means the result is similar to the state of the art under usual privacy requirements.

2. There is no comparison with Balle et al. when \sigma > 0.6. I am not sure whether the proposed algorithm is better or worse than Bella et al. in the standard range of noise (1 < \epsilon < 3).

3. I think this paper can be improved by providing an experiment about the accuracy of different models under different privacy guarantees/noise levels (I am not an expert on NLP, please let me know if some experiments in the paper already analyzed some metrics similar to accuracy)

4. I suggest the authors improve the readability by adding a statement that Renyi DP-SGD is implemented in the same way as DP-SGD. Otherwise, it's hard for the readers to know how Renyi DP is achieved for SGD.

5. I would also suggest discussing other applications for Renyi DP. For example, improving robustness.

**Summary Of The Paper:**

This paper provides an improved theoretical guarantee for the defense against reconstruction attacks using Renyi differential privacy. The correctness of the theoretical analysis is supported by the experimental analysis

**Summary Of The Review:**

I am not an expert on NLP, and all my judgments are based on the differential privacy part of this paper.

As mentioned in the Strength And Weaknesses, some details of this paper are not clear and there might be some technical issues. The presentation can also be improved.

---

> ### Author Response · Authors · 2022-11-18
> **Response to the review**
>
> We thank the reviewer for their review. We want to emphasize that our paper is the first to show the kind of protections that differential privacy provides when it comes to reconstruction of text data. Since the work of Carlini et al. 2021, training data extraction has become a major concern for language models and to the best of our knowledge, there is no work analyzing how DP protects specifically against these  attacks (despite general considerations about epsilon). We specifically tackle this issue, and in particular show that longer secrets are better protected which is both intuitive and not shown by any other privacy paper in the literature.
>
> > The theoretical bound in this paper is just slightly better than the state of the art when \sigma becomes larger (e.g., 0.5), which means the result is similar to the state of the art under usual privacy requirements.
>
> Our privacy analysis is twofold: first, we introduce a new way to measure privacy via the trade-off (leakage, probability of secret) rather than (epsilon, delta) which comes with many practical benefits, in particular the fact that longer secrets are more difficult to reconstruct (as a percentage of their length). The second part of our analysis is the improvement over the naive bound, which is quite significant for lower values of privacy (see also response below)..
>
> > There is no comparison with Balle et al. when \sigma > 0.6. I am not sure whether the proposed algorithm is better or worse than Bella et al. in the standard range of noise (1 < \epsilon < 3).
>
> We emphasize that our bound is *not* comparable to Balle et al. and provide the comparison to further the analogy from (epsilon, delta) to (leakage, probability of secret). That being said, we think the reviewer raises a valid point and we have run our bounds on different settings of DP-SGD and added them to the appendix (Figure 6). We see that in general, Mironov and Balle provide similar results, while our bound is better. Our bound really shines in the lower privacy regime (higher epsilons), and we believe it depends on epsilon but not on sigma (given that Figure 1 and 6 show comparable results with different sigmas but similar ranges of epsilons).
>
> > I think this paper can be improved by providing an experiment about the accuracy of different models under different privacy guarantees/noise levels (I am not an expert on NLP, please let me know if some experiments in the paper already analyzed some metrics similar to accuracy)
>
> Perplexity (or log-probability) is the standard measure of performance for language models, although the equivalent to “accuracy” would be something like top-10 error rate which is rarely used in NLP. We emphasize that our goal in the paper is not to explore the privacy-accuracy trade-offs of language models and that great progress has been achieved on this already by other works (Yu et al. 2021, Li et al. 2021). Rather, we study the empirical leakage on language models trained with and without differential privacy, and show that DP directly provides leakage protection.
>
> > I suggest the authors improve the readability by adding a statement that Renyi DP-SGD is implemented in the same way as DP-SGD. Otherwise, it's hard for the readers to know how Renyi DP is achieved for SGD.
>
> We will add a clarifying paragraph stating that DP-SGD provides Renyi DP accounting, which can be converted to traditional DP (epsilon, delta) or reconstruction guarantees as we show.
>
> > I would also suggest discussing other applications for Renyi DP. For example, improving robustness.
>
> We thank the reviewer for these suggestions, we will include it in the related work.

---

### Author Response · Authors · 2022-11-18
**Update of the paper**

We thank the reviewers for their review. We have updated the paper to include plots for other privacy parameters (Figure 6 in the Appendix), that show that our guarantees do not rely on specific values of sigma but display the same picture when looking at the same ranges of epsilon. In particular, our paper is the first to show that for privacy budget that are relatively high (epsilon > 8), training with RDP still provides a level of protection against extraction of rare secrets.

---

### Decision · Program_Chairs · 2023-01-20

**Decision:**

Reject

**Justification For Why Not Higher Score:**

The results are not strong enough to justify acceptance. The improvement is not significant compared to the results implied by prior work.

**Justification For Why Not Lower Score:**

N/A

**Metareview: Summary, Strengths And Weaknesses:**

The paper studies the protection guarantees of Renyi Differential Privacy (RDP) against reconstruction attacks. The authors derive an improved theoretical bound on the information leakage of RDP mechanisms compared to prior work [Mironov et al. 2019, Balle et al. 2020], hence, they show an improved protection against extraction of rare secrets from training data in low privacy regimes ($\epsilon > 8$). The authors provide empirical results on benchmark datasets to support their claims.

Despite the importance of this problem and the fact that the paper has some merits (deriving and improved theoretical bound and conducting a range of experiments on real-world datasets), the results are not strong enough to justify acceptance.The improvement is not significant compared to the results implied by prior work (particularly, [Balle et al. 2020]). According to the newly added empirical results (comparison to [Balle et al. 2020] and [Mironiov et al. 2019] added during the discussion phase), the improvement is only noticeable and meaningful in the (very) low privacy regime ($\epsilon$ is quite large) when the size of the secret is relatively large. This limits the significance of the results and makes them less compelling. Also, the added comparisons seem to be rushed (that was clear from the missing plot labels and the sparse discussion). The paper could benefit from an extensive, more careful empirical comparison to prior work. As it stands,  the claims of significance are not quite convincing.